# Cystic Kidney Diseases That Require a Differential Diagnosis from Autosomal Dominant Polycystic Kidney Disease (ADPKD)

**DOI:** 10.3390/jcm11216528

**Published:** 2022-11-03

**Authors:** Akinari Sekine, Sumi Hidaka, Tomofumi Moriyama, Yasuto Shikida, Keiji Shimazu, Eiji Ishikawa, Kiyotaka Uchiyama, Hiroshi Kataoka, Haruna Kawano, Mahiro Kurashige, Mai Sato, Tatsuya Suwabe, Shinya Nakatani, Tadashi Otsuka, Hirayasu Kai, Kan Katayama, Shiho Makabe, Shun Manabe, Wataru Shimabukuro, Koichi Nakanishi, Saori Nishio, Fumihiko Hattanda, Kazushige Hanaoka, Kenichiro Miura, Hiroki Hayashi, Junichi Hoshino, Ken Tsuchiya, Toshio Mochizuki, Shigeo Horie, Ichiei Narita, Satoru Muto

**Affiliations:** 1Nephrology Center, Toranomon Hospital, Tokyo 105-8470, Japan; 2Kidney Disease and Transplant Center, Shonan Kamakura General Hospital, Kanagawa 247-8533, Japan; 3Division of Nephrology, Department of Medicine, Kurume University School of Medicine, Fukuoka 830-0011, Japan; 4Department of Nephrology, Saiseikai Nakatsu Hospital, Osaka 530-0012, Japan; 5Department of Nephrology, Saiseikai Matsusaka General Hospital, Mie 515-8557, Japan; 6Department of Endocrinology, Metabolism and Nephrology, Keio University School of Medicine, Tokyo 160-8582, Japan; 7Department of Nephrology, Tokyo Women’s Medical University, Tokyo 162-8666, Japan; 8Department of Urology, Juntendo University Graduate School of Medicine, Tokyo 113-0033, Japan; 9Department of Advanced Informatics for Genetic Disease, Juntendo University Graduate School of Medicine, Tokyo 113-0033, Japan; 10Division of Kidney and Hypertension, Department of Internal Medicine, Jikei University School of Medicine, Tokyo 105-8461, Japan; 11Division of Nephrology and Rheumatology, National Center for Child Health and Development, Tokyo 157-8535, Japan; 12Department of Metabolism, Endocrinology and Molecular Medicine, Osaka Metropolitan University Graduate School of Medicine, Osaka 545-8585, Japan; 13Division of Clinical Nephrology and Rheumatology, Niigata University Graduate School of Medical and Dental Sciences, Niigata 951-8510, Japan; 14Department of Nephrology, Faculty of Medicine, University of Tsukuba, Ibaraki 305-8575, Japan; 15Department of Cardiology and Nephrology, Mie University Graduate School of Medicine, Mie 514-8507, Japan; 16Department of Child Health and Welfare (Pediatrics), Graduate School of Medicine, University of the Ryukyus, Okinawa 903-0215, Japan; 17Department of Rheumatology, Endocrinology and Nephrology, Faculty of Medicine and Graduate School of Medicine, Hokkaido University, Sapporo 060-8638, Japan; 18Department of General Internal Medicine, Daisan Hospital, Jikei University, School of Medicine, Tokyo 105-8471, Japan; 19Department of Pediatric Nephrology, Tokyo Women’s Medical University, Tokyo 162-8666, Japan; 20Department of Nephrology, Fujita Health University, Aichi 470-1192, Japan; 21Department of Blood Purification, Tokyo Women’s Medical University, Tokyo 162-8666, Japan; 22PKD Nephrology Clinic, Tokyo 104-0032, Japan; 23Department of Urology, Juntendo University Nerima Hospital, Tokyo 177-8521, Japan

**Keywords:** cystic kidney diseases, ADPKD, differential diagnosis, hereditary

## Abstract

Autosomal dominant polycystic kidney disease (ADPKD) is the most common hereditary cystic kidney disease, with patients often having a positive family history that is characterized by a similar phenotype. However, in atypical cases, particularly those in which family history is unclear, a differential diagnosis between ADPKD and other cystic kidney diseases is important. When diagnosing ADPKD, cystic kidney diseases that can easily be excluded using clinical information include: multiple simple renal cysts, acquired cystic kidney disease (ACKD), multilocular renal cyst/multilocular cystic nephroma/polycystic nephroma, multicystic kidney/multicystic dysplastic kidney (MCDK), and unilateral renal cystic disease (URCD). However, there are other cystic kidney diseases that usually require genetic testing, or another means of supplementing clinical information to enable a differential diagnosis of ADPKD. These include autosomal recessive polycystic kidney disease (ARPKD), autosomal dominant tubulointerstitial kidney disease (ADTKD), nephronophthisis (NPH), oral-facial-digital (OFD) syndrome type 1, and neoplastic cystic kidney disease, such as tuberous sclerosis (TSC) and Von Hippel-Lindau (VHL) syndrome. To help physicians evaluate cystic kidney diseases, this article provides a review of cystic kidney diseases for which a differential diagnosis is required for ADPKD.

## 1. Introduction

Autosomal dominant polycystic kidney disease (ADPKD) is the most common form of hereditary cystic kidney disease, affecting one in 1000–2500 individuals and is mainly caused by mutations in the *PKD1* (78%) and *PKD2* (15%) genes [1]. Approximately 7% of patients with ADPKD appear not to have a *PKD1*/*2* mutation [2,3]. Among ADPKD patients, the severity of the effects of these mutations on renal survival was reported to be as follows: *PKD1* truncating mutation > *PKD1* non-truncating mutation > *PKD2* mutation > no *PKD1* or *PKD2* mutation [4,5]. Proliferative cysts in both kidneys increase in number and size as kidney function impairment advances, often introducing complications such as hypertension, multiple hepatic cysts, cerebral aneurysm, heart valve disease, and colonic diverticula. Diagnosis often occurs in adulthood when symptoms are present, but a differential diagnosis of ADPKD is required to distinguish it from other cystic kidney diseases, that are also characterized by renal cysts. Patients with ADPKD often have a family history of this disease (75–90% having a parent with ADPKD [1]). However, among patients with no family history (the 10–25% who have no parent with ADPKD [1]), some 40% reportedly have no observed mutation in the *PKD1* or *PKD2* genes [6,7]; this suggests that careful diagnosis is required to differentiate between ADPKD and other cystic kidney diseases. The only therapeutic drug for patients with ADPKD is tolvaptan, a vasopressin V2-receptor antagonist that slows the rate of decline of the estimated glomerular filtration rate (eGFR) [8,9,10]. Accordingly, it is important to diagnose ADPKD from other cystic kidney diseases. To help physicians evaluate cystic kidney diseases, this article provides a review of the cystic kidney diseases for which a differential diagnosis is required for ADPKD (Table 1).

## 2. Cystic Kidney Diseases That Need to Be Excluded from an ADPKD Diagnosis

### 2.1. Multiple Simple Renal Cysts

Simple renal cysts are an acquired cystic kidney disease which primarily occur in the renal cortex; they present as thin-walled, saclike structures in the renal cortex and renal pelvic environment. Usually a simple cyst of only unilateral, but with age the number of cysts increases and the frequency with which they are present in both kidneys increases [11]. Simple renal cysts are rarely found in childhood but increase with age after the age of 30. Autopsy reports indicate the presence of at least one renal cyst in 50% of patients aged 50 years or above. Prevalence rates can vary depending on age, gender, and ethnicity [12,13,14].

Current assumptions regarding renal cyst formation indicate abnormal hypertrophic response in nephrons due to renal ischemia or obstruction, with cyst growth and compensatory hyperfiltration leading to nephron loss [14]. As renal cysts have a slow growth rate, many years may pass between onset and detection of simple renal cysts. The risk of latent kidney function impairment increases with age, and aging is associated with decreased glomerular filtration rate and increased incidence of renal cysts [15]. Simple renal cysts are normally asymptomatic (unless they are exceedingly large) and have little impact on kidney function. Typically, they increase slowly, or remain unchanged over time [11]. Symptoms include hemorrhagic cysts (2–4% of cases), infection, and rupture. Risk factors include aging, male gender, and hypertension.

Ultrasonograms show that simple renal cysts in the renal cortex and renal pelvic environment are round, anechoic, and accompanied by posterior echo enhancement [16]. Some cysts project beyond the renal parenchyma, with cysts located in the renal pelvic environment known as parapelvic renal cysts; these may easily be mistaken for hydronephrosis at first glance. Hydronephrosis and simple renal cysts can be differentiated by confirming individually isolated cysts one by one. Simple renal cysts are designated as Class I through computed tomography (CT) imaging using the Bosniak classification (Table 2) [17]. CT values are scaled from −10 to +20 Hounsfield units (HU), in the same manner as water, with smooth and sharp boundaries [16]. Contrast-enhanced CT imaging shows a uniform image with absolutely no contrast (Figure 1). Magnetic resonance imaging (MRI) displays the same signaling as water, with a T2-weighted image showing uniform hyperintensity (high-signal intensity) and a T1-weighted image exhibiting hypointensity (low-signal intensity), with no contrast whatsoever following contrast imaging. Fluid hemorrhage from the cyst interior, or the high-protein nature may appear as high absorbance on non-contrast CT imaging, and wall thickening, septum partitioning, and calcification may be observed in some cysts, giving them the name “complex cysts.” The Bosniak classification (Table 2) is helpful in the differentiation of malignant cysts; this is of utmost importance in diagnosing atypical renal cysts [17].

### 2.2. Acquired Cystic Kidney Disease (ACKD)

In long-term dialysis patients, the renal parenchyma in both kidneys may show atrophy and proliferation of microcysts. This condition is known as acquired cystic kidney disease (ACKD), and cancer may develop from these cyst walls. It has also been shown that the longer patients continue with dialysis, the greater the incidence of ACKD. Truong et al. reported that about 50% of dialysis patients developed ACKD, citing an incidence rate by dialysis interval of 13% for 2 years, 50% for 6 years, 87% for 9 years, and almost 100% for patients on dialysis for 10 years or more [18]. ACKD also appears in patients with chronic kidney disease (CKD); a report by Choyke et al. (2000) highlighted that ACKD was observed in 8–13% of patients with CKD prior to initiation of dialysis [19].

The developmental mechanism for ACKD remains largely unclarified. Grantham (1991) hypothesized that uremic toxic substances and electrolyte imbalance, factors specific to kidney failure, act as stimulants, with hyperplasia developing from compensatory hypertrophy of the tubular epithelium, resulting in cyst formation [20]. Other renal factors such as ischemic change, inflammation, hormonal abnormality, and precipitation of oxalic acid and other salts may also play a role [20].

ACKD is asymptomatic, but screening is preferable as hemorrhage and renal cancer are likely complications. Ultrasound is the most common form of screening as a lower cost and non-invasive procedure. Ultrasonograms can highlight whether the kidney is atrophied or a normal size, if a thin cortex is producing high echogenicity, whether three or more cysts are present, and if it is able to detect these three symptoms in both kidneys. The absence of kidney enlargement in ACKD greatly differentiates it from ADPKD [21]. Meanwhile, CT imaging can indicate the presence of smaller cysts. Non-contrast CT images typically show proliferation of low-absorption lesions in the bilateral renal cortex, but without contrast medium it can be difficult to differentiate cysts from solid lesions. Contrast enhancement cannot characteristically be seen for cysts in contrast CT images. Imaging with MRI does, however, allow for the sharpest presentation of cysts. T1-weighted images display hypointensity and T2-weighted images, hyperintensity. In renal cancer linked to ACKD, a solid intracystic composition can often be verified by CT and MRI images. Ishikawa et al. (2011) reported the usefulness of ultrasonograms employing contrast medium in viewing intratumoral blood flow to simplify diagnosis [22]. Furthermore, Kitajima et al. (2019) reported that fluorodeoxyglucose-positron emission tomography (FDG-PET)/CT images were useful in detecting increased FDG accumulation in some ACKD-related cases [23]. The Bosniak classification is utilized for differentiation between benign and malignant cystic kidney disease (Table 2) [17]. Contrast CT images for Bosniak Class IV cases are shown in Figure 2; cystic lesions on the left renal inferior pole show a solid densely stained area indicating a 90% or greater chance of malignant progression, so surgery is indicated in this case.

Additionally, ACKD cysts are known to shrink in size following kidney transplant [24]. However, one should note that the risk of renal cancer remains [25,26,27].

### 2.3. Multilocular Renal Cyst (Also Referred to as Multilocular Cystic Nephroma or Polycystic Nephroma)

Multilocular renal cyst is a rare form of benign cystic kidney disease (neoplasm) which is also referred to as multilocular cystic nephroma or polycystic nephroma. Over 200 cases of this kidney cystic adenoma have been reported since the disease was first documented in 1892 [28,29,30]. Age distribution is bimodal (2 to 4 years, and patients in their forties to sixties). In patients under the age of 4 years, the ratio of males to females is 3:1, but this ratio is 1:9 in adults, with female patients predominating [29,30]. While sex hormones have been reported to play a role in this disease [31], its causality remains unclear. These lesions are noncommunicative with the renal pelvis; they are multilocular cyst lesions with no nephron structure and are largely supported by epithelial cells. Kidney tissue is normal in areas where lesions are not present.

Patients may exhibit nonspecific symptoms such as abdominal pain, hematuria, and urinary tract infections [29,30]. Ultrasonograms show alternating hypoechoic and hyperechoic cysts, with Doppler ultrasonograms helping to differentiate between benign and malignant cysts. CT imaging can verify well-defined borders of polycystic tumors (Figure 3). Calcification is rare but may be seen within the septum or on the cyst wall, complicating the differentiation from a malignant tumor. MRI imaging usually shows hypointensity on T1-weighted images and hyperintensity on T2-weighted images, with a hypointensive septum due to inclusion of tissue components (Figure 3).

As already noted, ultrasound, CT, and MRI imaging are important in tumor identification; however, it is difficult to reach a differential diagnosis between polycystic kidney disease and renal cell carcinoma based on clinical information and imaging results alone. A definitive diagnosis requires surgical treatment in the form of a unilateral nephrectomy, or pathological evaluation following a partial nephrectomy [29,30].

### 2.4. Multicystic Kidney/Multicystic Dysplastic Kidney (MCDK)

Multicystic kidney/multicystic dysplastic kidney is referred to as multicystic dysplastic kidney (MCDK) and is a cystic kidney disease expressed during the fetal stage of development as a result of the unsuccessful formation of nephrons and collecting ducts. This disease is believed to have an incidence rate of 1 in 1000–4300 individuals [32], with 70% of cases being unilateral. Most cases are diagnosed through fetal ultrasound imaging [33], and spontaneous regression occurs in 24% of cases by the age of 5 years [34], and in over 50% of cases by the age of 10 years [34,35]. Severe kidney function impairment occurs in bilateral MCDK cases, and this requires continuous observation as there is a possibility it may lead to oliguria or amenorrhea [36].

Imaging findings include a large aggregation of variously sized and mutually noncommunicative botryoidal cysts (Figure 4a). Renal dynamic scintigraphy shows a decreasing glomerular filtration rate (Figure 4b), with compensatory contralateral hypertrophy [37]. Contralateral urinary tract complications occur with a 30% frequency, the most common being vesicoureteral reflux (VUR) [38]. Renal prognosis is favorable when no contralateral complications such as VUR are observed, but complications do introduce the risk of progressive kidney function impairment [39]. MCDK closely resembles occlusive nephropathy resulting from severe urinary tract obstruction, but the latter is differentiated by its tendency to progress over time. Other hereditary cystic kidney diseases are characterized by cysts found on the medulla or entire cortex, and have a functional renal parenchyma, whereas the renal parenchyma in MCDK is non-functional [36]. MCDK was thought to be a nonhereditary renal disease, but in recent years, a human *HNF1B* gene mutation was found to have a causal relationship with a renal developmental defect accounting for 10–30% of all cases of kidney and urinary tract congenital abnormalities (CAKUT) [40], MCDK, and autosomal dominant tubulointerstitial kidney disease (ADTKD) [41]; 1 in 10 patients with unilateral MCDK are reported to have an *HNF1B* gene mutation [42].

### 2.5. Unilateral Renal Cystic Disease (URCD)

Unilateral renal cystic disease (URCD), named by Levine et al. in 1989 [43] presents with cysts, thereby resembling ADPKD in imaging and histological findings (Figure 5) [44]. Although, it does differ from ADPKD in an absence of family history, and failure to progress to end-stage kidney disease (ESKD) [45]. Differential diagnosis is also helped by the fact that URCD has no extrarenal complications, such as hepatic cysts or cerebral aneurysm [46]. Diseases with unilateral or other cysts resembling those of URCD require a differential diagnosis; these include: multicystic dysplastic kidney, multilocular renal cysts, uteropelvic junction obstruction, non-obstructive megaureter, and simple renal cysts [46]. Differential diagnosis may also be required on the rare occasion for patients with ADPKD who have a *PKD1* gene mutation and aplastic or hypoplastic kidney, or uteropelvic junction obstruction [47]. Ongoing observation is important as there are reports of unilateral cystic kidney diseases progressing over time to the bilateral stage during childhood [48].

## 3. Cystic Kidney Diseases That Require a Differential Diagnosis from ADPKD

### 3.1. Autosomal Recessive Polycystic Kidney Disease (ARPKD)

Autosomal recessive polycystic kidney disease (ARPKD) presents as renal cystic lesions reflecting the expansion of the collecting duct with kidney function impairment, and congenital hepatic fibrosis characterized by bile duct dysplasia and intrahepatic periportal fibrogenesis [49]. The severity of these hepatic and renal lesions varies greatly from one individual to another, but it is reported to occur in one among every 8000 to 40,000 individuals [49,50,51]. Although many lesions appear in the perinatal stage or in childhood [49], they can manifest at any age, including well into adulthood. In an international ARPKD cohort study (n = 470 patients), 45 adult patients were evaluated in a cross-sectional analysis (median age, 19.9 years; average age, 21.4 years). The majority of these patients showed multiple bilateral cysts (≥10 cysts per kidney) on ultrasound, with half cases reported to be CKD stage 1–3. These patients required examination for conditions such as thrombocytopenia and splenomegaly which are derived from hepatic cysts, as well as for portal hypertension, to enable a differential diagnosis of ADPKD [52].

The *PKHD1* gene (67 exons) is the causative gene and is found on 6p21.1-p12. It has a coding protein comprised of 4074 amino acids and is defined as a receptor-like protein which passes once through a cellular membrane described as fibrocystic or polyductin [53,54]. This disease has an autosomal recessive form, with 80% being homozygous or compound heterozygous, and 95% or over having one or more *PKHD1* gene mutations [50,55,56,57,58]. Biallelic or truncated mutations are the most serious and usually result in embryonic or neonatal stage death, while one or more missense mutations are deemed relatively mild [55,59,60]. A recent study that analyzed *PKHD1* gene missense mutations (taking amino acid location into account) in 304 patients with ARPKD (representing 277 family lineages) found less likelihood for progression to kidney failure in individuals with a missense gene mutation at amino acids 709–1837, but portal hypertension and severe hepatic complications in those with a missense gene mutation at amino acids 2625–4074 [61]. While patients within the same family lineage usually follow the same clinical course, varying levels of disease severity have been reported in about 20% of family lineages among patients within the same family lineage [50]. This suggests the possible effect, not only of the *PKHD1* gene mutation, but also of environmental factors and other modified genes [59].

The second causative gene is *DZIP1L*, found on 3q22.1-q23 [62]. The phenotype caused by this gene mutation appears in the embryonic or early post-natal stage but is described as moderate ARPKD [62,63] as it does not lead to neonatal death.

As there are no established diagnostic criteria, genetic testing should be considered for cases in which diagnosis solely through clinical conditions and imaging findings is difficult [52]. Since small cystic lesions, which are mainly dilated renal tubules rather than cysts, are diffusely present [64], the entire kidney becomes hyperechoic rather than lumpy and hypoechoic [64]. In the liver, congenital hepatic fibrosis, including hepatic swelling may be present. Typically, CT imaging reflects kidney enlargement with small embedded cysts or a striated appearance with collecting duct enlargement [64] (Figure 6). CT imaging may also be used to detect portal hypertension which is associated with congenital hepatic fibrosis. MRI imaging typically shows hypointensity on T1-weighted images and hyperintensity on T2-weighted images which is similar to ADPKD findings, but ARPKD has fewer complicated cysts associated with hemorrhaging and infection [64] (Figure 6).

As no established disease-specific treatment exists, patients with ARPKD are usually treated individually with symptomatic therapy. Since most patients experience hypertension, proactive antihypertensive therapy is important [52]. Kidney replacement therapy is appropriate for patients with progression of kidney function impairment, and other therapies, including hemodialysis, peritoneal dialysis, and kidney transplantation may also be considered. Hepatic complications such as cholangitis and esophageal varices are also possible.

### 3.2. Autosomal Dominant Tubulointerstitial Kidney Disease (ADTKD)

Autosomal dominant tubulointerstitial kidney disease (ADTKD) is a general term for a group of diseases inherited in an autosomal dominant manner and characterized by tubulointerstitial fibrogenesis and progressive kidney function impairment. First proposed in the 2015 Kidney Disease: Improving Global Outcomes (KDIGO) consensus report [41], ADTKD is currently believed to be caused by gene mutations *UMOD*, *REN*, *MUC1*, *HNF1B*, *SEC61A*, and *DNAJB11* [65,66,67]. This disease has previously been referred to by various terms, such as familial juvenile hyperuricemic nephropathy and medullary cystic kidney disease, but with gene identification for ADTKD now available, diagnosis has become unified. For patients with kidney failure (adolescence through to old age), with a family history of kidney function impairment, who show few findings on urinary examination, and experience a slow decline in kidney function, ADTKD should be considered. As ADTKD often occurs in patients of advanced age, it is often mistaken for nephrosclerosis, or other renal diseases, and can be overlooked. Since ADTKD symptoms are nonspecific, genetic testing is required for an accurate diagnosis. Accordingly, this disease is often referred to as “ADTKD- (abnormal gene name).” *HNF1B* aside, the five genetic abnormalities often manifest when abnormal protein accumulates within the cell, bringing about endoplasmic reticulum stress which leads to cell dysfunction and tubulointerstitial damage [65].

ADTKD-*UMOD* appears as a mutation of the *UMOD* gene found on chromosome 16p12.3 [41]. The protein made by *UMOD* is uromodulin, also known as the Tamm–Horsfall protein, which is produced in kidney epithelial cells by the thick ascending limb (TAL) of the loop of Henle and secreted in urine. Physiologically, uromodulin is the most abundant protein excreted in urine and is the main component of urinary casts [67]. It is greatly glycosylated and enables protein polymerization. Uromodulin functions in TAL/distal ureter electrolyte transport safeguards against urinary tract infection and calcium-containing nephrolithiasis, and aids in topical and systemic immunoregulation. TAL dysfunction, which includes downregulation of Na^+^-K^+^-2Cl cotransporter, has been proven in ADTKD-*UMOD* [68].

Hyperuricemia is observed from childhood, and the mechanism is believed to include obstructed transitions in the luminal surface of the Na^+^-K^+^-2Cl cotransporter; this causes a slight fluid volume decrease and the compensatory appearance of a uric acid transporter in the secondary proximal renal tubule, along with an increase in uric acid [69]. Clinical observations include hyperuricemia, gouty arthritis, a decline in urine concentrating ability, and progressive kidney function impairment. Patients are reported to reach ESKD at a median (range) age of 46 (39–57) years. Histologically, kidney atrophy is due to stroma fibrogenesis and nephron atrophy, microcyst formation has also been noted. Renal size ceases to be normal due to atrophy, with 25% cyst formation (Figure 7) [70].

ADTKD-*MUC1* occurs due to a mutation in the *MUC1* gene found on chromosome 1q22 [41]. Mutations in *MUC1* are difficult to identify through exon analysis using a next-generation sequencer (based on polymerase chain reaction [PCR]) as the mutation often contains one base insertion of cytosine in the GC-rich and PCR-resistant variable number tandem repeat (VNTR) area [71]. Mucin1 is the protein encoded by *MUC1* and is expressed in TAL, distal tubule, lung, small intestine, and stomach. Mucin1 safeguards the luminal surface of epithelial cells, and is a transmembrane glycoprotein involved in intracellular signaling and cell interaction. The mucin1 protein has been shown to activate cytoprotective hypoxia-inducible factor-1 (HIF-1) and the beta-catenin-dependent route in a mouse model of renal ischemic reperfusion damage [72]. It also activates the transient receptor potential cation channel, subfamily V, member 5 (TRPV5), a Ca2^+^ channel in the distal tubule, and acts to increase urine Ca2^+^ resorption [67]; this suggests that there is less mucin1 protein excretion in the urine of patients with renal calculus than in healthy individuals [72]. Mutant mucin1 protein influences endoplasmic reticulum stress and tubulointerstitial nephritis which causes cell dysfunction. Clinical observations include progressive kidney function impairment, with patients reaching ESKD at an average age of 45 years [73]. The only noted extrarenal disease is gout, which appears comparatively infrequently with ADTKD-*UMOD* [73]. Imaging reveals no kidney enlargement; rather, the kidney tends to show atrophy. Cyst formation has been reported in 37% of ADTKD-*MUC1* patients [74]; however, not in the medulla. Histological findings show stroma fibrogenesis, nephron atrophy, and microcyst formation.

ADTKD-*REN* occurs due to a mutation in the *REN* gene found on chromosome 1q32.1 [41]. The disease is extremely rare, but in recent years, genetic and clinical spectra have been further clarified [75]. Pre-prorenin is the protein encoded by *REN*; it is primarily comprised of 406 amino acids expressed on granule cells of the juxtaglomerular apparatus. The N-terminal signal peptide is removed by the endoplasmic reticulum through cellular processing, allowing the formation of prorenin. Prorenin ultimately becomes renin, which is comprised of 340 amino acids, is stored in vesicles, enabling regulated exocytosis. As a result of the impaired translocation of the endoplasmic reticulum and mild endoplasmic reticulum stress caused by the signal peptide, the heterozygous *REN* mutation causes deteriorated synthesis of prorenin and renin [75]. This signal peptide and pro-segment mutation frequently occurs, and it is indicated at an extremely early age (about half of cases presenting before 10 years), in slight hyperkalemia, anemia, acidosis, slowly progressive CKD with a median kidney lifespan of 60 years, and early-onset gout. On the other hand, a mutation in part of the mature renin protein presents in adult-onset CKD and gout, with patients reaching ESKD in their late 60s. Renal biopsy specimens show deteriorating renin production [75], with signal peptide mutant non-glycosylated pre-prorenin reportedly accumulating in the cytoplasm [76]. When a mutation occurs in the mature renin protein, the mutant protein accumulates in the endoplasmic reticulum, activating endoplasmic reticulum stress and an unfolded protein response (UPR) [77]. As decreased renin activity is accompanied by low blood pressure and hyperkalemia, treatment is required and includes fludrocortisone. As sodium depletion also accompanies decreased renin activity, diuretics, a sodium-restricted diet, nonsteroidal anti-inflammatory drugs (NSAIDs) and RAAS-suppressing drugs should be avoided [65,76].

ADTKD-*HNF1B* occurs due to a mutation in the *HNF1B* gene found on chromosome 17q12 [41]. *HNF1B* encodes hepatocyte nuclear factor 1 B (HNF1B), a transcription factor which suppresses multiple genes expressed in the kidney, pancreas, and liver. As the HNF1B protein is involved in the transcription of various genes, congenital anomalies of the kidney and urinary tract, such as dysplastic kidney and renal aplasia, as well as renal cysts can be found in the kidney (Figure 8). Extrarenal diseases include juvenile-onset diabetes mellitus and liver function impairment, as well as abnormalities of the reproductive organs [67]. HNF1B is also involved in the activation of *PKHD1*, *PKD2*, and *UMOD* gene transcription factors which are linked to cystic disease. As HNF1B regulates transcription of the *FXYD2* gene which encodes the gamma subunit of Na^+^-K^+^-ATPase expressed in distal tubules, it also impacts electrolyte regulation. Screening for extrarenal manifestations such as diabetes and liver function impairment is important.

The hetero-missense mutation *SEC61A1* which encodes the alpha 1 subunit of SEC61 translocon is also linked to ADTKD [67]. As this subunit constitutes a part of the translocon which transports newly synthesized protein to the endoplasmic reticulum, when the SEC61 translocon structure mutates, there is a disruption to the post-translational modification, folding, and sorting of the transmembrane protein, resulting in endoplasmic reticulum stress. Clinically, this disease is characterized as early stage, it is slowly progressive with CKD and gout occurring at an early age. Ultrasound testing reveals a normal or small size kidney, and bilateral polycystic kidneys. Renal biopsy shows tubular atrophy and interstitial fibrosis. Extrarenal symptoms include congenital anemia, neutropenia, and hypogammaglobulinemia. Growth retardation, cleft palate, spina bifida, mild cognitive impairment, and polydactyly may also be evident [66].

ADTKD-*DNAJB11* is also recognized as a form of ADTKD [67]. DNAJB11 is a cofactor of BiP (GRP78), an endoplasmic reticulum chaperone which suppresses protein folding, transport, and degradation. It was recently clarified that a *DNAJB11* heterozygous mutation is linked to a clinical phenotype expressed in a hybrid between ADTKD and ADPKD [78]. Patients typically transition to ESKD at a median age of 75 years, with a favorable renal prognosis compared to ADTKD-*UMOD* and ADTKD-*MUC1* [79]. This disease is characterized by bilateral, multiple small renal cysts without kidney enlargement, and by hepatic cysts in 50% of all cases. Renal biopsies in several cases show tubulointerstitial fibrosis. Early-onset gout is not reported in these patients, but circulatory abnormalities such as intracranial aneurysm, thoracic aorta enlargement, and carotid artery dissociation have been indicated [78,79].

While six genetic abnormalities associated with ADTKD have been described here, new gene mutations are also likely to be found in patients showing histopathological findings suggesting ADTKD.

### 3.3. Nephronophthisis (NPH)

Nephronophthisis (NPH) is characterized by impaired urinary concentration, chronic tubulointerstitial nephritis, renal cystic lesions, and accompanying kidney function impairment and progress to ESKD by the age of 30 years [80]. NPH accounts for 15% of all diseases in which ESKD is reached by the age of 30 years, and it occurs in 1 or 2 out of every 100,000 individuals [81]. Clinically, there are three subtypes classified according to the median age at which ESKD is reached: infantile NPH (median age 1 year), juvenile NPH (median age 13 years), and adolescent/adult NPH (median age 19 years) [80,82,83], with patients reported to reach ESKD between 20 and 60 years of age [80,84,85]. This disease is autosomal recessive with 25 known causative genes: *NPHP1*, *NPHP2*/*INVS*, *NPHP3*, *NPHP4*, *NPHP5*/*QCB1*, *NPHP6*/*CEP290*, *NPHP7*/*GLIS2*, *NPHP8*/*RPGRIP1L*/*MKS5*, *NPHP9*/*NEK8*, *NPHP10*/*SDCCAG8*/*SLSN7*, *NPHP11*/*TMEM67*/*MKS3*, *NPHP12*/*TTC21B*/*JBTS11*, *NPHP13*/*WDR19*, *NPHP14*/*ZNF423*, *NPHP15*/*CEP164*, *NPHP16*/*ANKS6*, *NPHP17*/*IFT172*, *NPHP18*/*CEP83*, *NPHP19*/*DCDC2*, *NPHP20*/*MAPKBP1*, *NPHP1L*/*XPNPEP3*, *NPHP2L*/*SLC41A1*, *TRAF3IP1*, *AH11*/*JBTS3*, *CC2D2A*/*MKS6* [81]. The *NPHP1* gene is responsible for the most occurrences of NPH (20%), with the remaining genes each accounting for less than 1%. Two-thirds of all NPH cases have no clearly identifiable gene mutation [80]. Causative genes that are representative of adolescent/adult NPH include: *NPHP3*, *NPHP4*, and *NPHP9*/*NEK8,* but some cases of NPH do not clearly match up with any of these genes [80,81,86,87,88,89,90].

Clinical conditions include polyuria and polydipsia indicating an impairment to concentrate urine, hyponatremia accompanied by impaired sodium absorption, and symptoms characteristic to the progression of kidney function impairment. Typically, blood pressure is normal. There are distinct ultrasound findings, such as high luminance, normal to a slightly smaller sized kidney, and corticomedullary junction cysts, as well as the disappearance of the corticomedullary junction [64]. CT imaging shows renal parenchymal fibrogenesis, either a normal kidney size, or comparatively slightly smaller than average kidney size for the age of the patient, as well as cysts in the renal medulla and corticomedullary junction [64]. Some patients also show kidney enlargement resembling that of ARPKD [63] (Figure 9a), and MRI T2-weighted images show multiple cysts with hyperintensity [64] (Figure 9b). Pathological findings in the kidney include corticomedullary junction cysts, irregular hypertrophy, and overlapping of the tubular basement membranes, as well as tubulointerstitial nephritis [91]. Extrarenal lesions are reportedly found in 10–20% of patients, together with retinitis pigmentosa, cerebellar vermis hypoplasia, gaze palsy, hepatic fibrosis, and skeletal abnormalities. The following NPH-related diseases are based on the type of extrarenal lesions the patient presents with: Senior-Loken syndrome, Arima syndrome, Alstrom syndrome, RHYNS, Joubert syndrome (JS), Cogan syndrome, COACH, Meckel-Gruber syndrome (MKS), Boichis syndrome, Mainzer-Saldino syndrome, Bardet-Biedel syndrome, Ellis van Creveld syndrome, Jeune syndrome, and Sensenbrenner syndrome. In some patients, a different variant of the same gene produces a different phenotype and a different syndrome. For example, a nonsense mutation of the *CC2D2A* gene is associated with MKS, while a missense mutation is associated with JS. Similarly, the *TMEM231* gene is linked to MKS, OFD, and JS.

A diagnosis is based on clinical characteristics and confirmed by genetic testing [80,83,92]. However, as two-thirds of all cases have no clear gene mutation, a renal biopsy may be considered for a definitive diagnosis. In a report from Japan, pathological findings in the kidneys of older patients revealed a redundant thick tubular basement membrane characteristic of NPH, but genetic testing found no pathogenic mutation in the genes responsible [92].

No established disease-specific therapy exists, so symptomatic therapy is typical for individual patients [91]. Kidney replacement therapy is required if kidney function impairment progresses, and hemodialysis, peritoneal dialysis, or kidney transplant may be considered. Kidney transplantation results are reported to be comparatively favorable [93].

### 3.4. Multiple Abnormalities Accompanying Cystic Kidney Disease (OFD1, NPHP/SLS, JSRD, MKS, BBS, OFDS)

Cystic kidney diseases accompanied by multiple abnormalities include nephronophthisis (NPHP)/Senior-Loken syndrome (SLS), Joubert syndrome and related diseases (JSRD), Meckel syndrome (MKS), Bardet-Biedl syndrome (BBS), and orofacial digital syndrome (OFDS) [94]. Oral-facial-digital syndrome type 1 (OFD1), a type of orofacial digital syndrome, clinically manifests in adulthood and is a disease requiring differentiation from ADPKD.

### 3.5. Oral-Facial-Digital Syndrome (OFD) Type 1

Oral-facial-digital syndrome (OFD syndrome) is a hereditary cystic kidney disease which incorporates a variety of morphological abnormalities in the oral, facial, and finger and toe areas. Thirteen types of OFD syndrome are recognized: Type 1, Papillon-League-Psaume syndrome; type 2, Mohr syndrome; type 3, Sugarman syndrome; type 4, Baraiter-Burn syndrome; type 5, Thurston syndrome; type 6, Varadi-Papp syndrome; type 7, Whelan syndrome; type 8, Edwards syndrome; type 9, Gurrier syndrome; type 10, Figuera syndrome; type 11, Gabrielli syndrome; type 12, Moran-Barroso syndrome; and type 13, Degner syndrome [95]. The most common, OFD syndrome type 1, has an X chromosome-linked hereditary form, and as it has male lethality, the phenotype is found only in females. This disease appears in 1 in 50,000–250,000 individuals [96], with about 75% of cases being sporadic [97]. It is caused by an *OFD1* gene mutation on X chromosome Xp22 [98]. OFD1 encoded by the *OFD1* gene is found in centrioles and basal bodies and functions in the first stage of ciliogenesis. The *OFD1* gene mutation is reportedly related to four syndromes with X chromosome-linked recessive inheritance (Joubert syndrome, retinitis pigmentosa, primary ciliary dyskinesia, and Simpson-Golabi-Behmel syndrome), which have phenotypes observed in both males and females [99].

The renal lesions are usually viewed as polycystic kidney and typically appear in adulthood (20s and 30s) [100]. Kidney function impairment can be seen from the post-natal period, but usually progresses to kidney failure in adulthood once polycystic kidney becomes apparent. Taking all age groups into account, polycystic kidney occurrence rate is 37.3%, but in patients aged 18 years and older, it is reported to be 63% [100]. Pathologically, most of the cysts are glomerulus in derivation [101]. CT and MRI imaging show kidney size to be normal to slightly large, with no change in appearance due to cysts (Figure 10). Oral lesions may result in tongue malformation (lobulation, hamartoma, split), cleft palate/high-arched palate, and tooth malformation [100]. Cephalofacial lesions typically include symptoms such as down slanted palpebral fissures, nostril underdevelopment, telecanthus, retrognathia, flat face, frontal prominence, harelip, milia (which disappears by the age of 3 years), dry and brittle hair, and hair loss [100]. Skeletal lesions which frequently manifest include brachydactyly, clinodactyly, preaxial polydactyly, broad thumb, broad hallux, and morphological defects in the finger and toe. Short stature has also been linked to this disease [100]. Associated central nervous system lesions include deformities (colpocephaly, arachnoid cysts, porencephaly, gray matter heterotopia, cerebellum malformation, abnormal gyrus, microcephaly), mental retardation, cognitive disorders, and other deformities [100]. Other symptoms include deafness and the presence of cysts on the pancreas, liver, and ovaries [100]. As this is an X chromosome-linked hereditary disease, varying degrees of X chromosome inactivity may produce differing intrafamilial/interfamilial phenotypes.

Most oral-facial-digital deformities are diagnosed in the post-natal stage, but mild cases may not be diagnosed until the appearance of renal lesions. Renal cysts, central nervous system lesions, and skeletal deformities are also identified in other forms of ciliopathy, but tooth deformities are distinctive. Female patients suspected of having OFD1 type 1 syndrome may benefit from genetic testing for the *OFD1* gene.

As established disease-specific treatment does not exist, medical care is generally provided according to individual patient symptoms. Reconstructive surgery is one option for treatable symptoms. Kidney replacement therapy may be required if kidney function impairment progresses, and hemodialysis, peritoneal dialysis, and kidney transplant may also be considered.

### 3.6. Neoplastic Cystic Kidney Diseases (Tuberous Sclerosis, Von Hippel-Lindau Syndrome)

#### 3.6.1. Tuberous Sclerosis (TSC)

Tuberous sclerosis (TSC) is a hereditary systemic disease characterized by systemic hamartoma, with *TSC1* and *TSC2* identified as the genes responsible for this condition. Hamartin–Tuberin complex, a *TSC1* and *TSC2* gene product, is involved in cell proliferation through mTOR inhibition, with resulting hamartoma formation in the skin, nervous system, kidney, lung, bone, and elsewhere, together with abnormalities in the *TSC1* and *TSC2* genes [102,103,104]. The disease is inherited in an autosomal dominant manner and occurs in one in 10,000 individuals.

New diagnostic criteria were proposed at the second TSC Clinical Consensus Conference in 2012 and are summarized in Table 3. Criteria for genetic diagnosis were added at this time, and criteria for clinical diagnosis were modified [105].

Distinctive renal lesions associated with tuberous sclerosis include angiomyolipoma (AML)/renal cysts/renal cell carcinoma. A study by Wataya-Kaneda et al. (2013) found that 71% of 166 Japanese patients with tuberous sclerosis had renal lesions (61% AML, 28% renal cysts, 2.6% renal cancer) [106]. Renal lesions are reportedly the primary organ in which lesions determine the prognosis of TSC in patients aged 10 years and above [107]. Only 1% of patients with TSC require kidney replacement therapy; this is due to the high mortality rate by the age of 20 years [108]. In the proposed international diagnostic criteria (Table 3), renal AML is listed as one of the major features, while multiple renal cysts are noted as a minor feature (Figure 11) [105]. It is estimated that about 30% of TSC patients do not show typical clinical symptoms other than renal cysts, and it is thought that there are a certain number of cases that have been misdiagnosed as ADPKD [109].

Renal cysts complicated by TSC arise from various places of nephrons such as glomerulus, and kidney function deteriorates due to the frequent occurrence and increase in renal cysts [110]. mTOR inhibitors are recommended for renal AML [105], and while no established therapy exists for renal cysts, Siroky et al. (2017) utilized mTOR inhibitors in 15 tuberous sclerosis patients with renal cysts and reported a 71% decrease in kidney volume [111]. As *TSC2*, the gene responsible for tuberous sclerosis, is adjacent to *PKD1*, the gene responsible for ADPKD, *TSC2/PKD1* contiguous gene deletion syndrome is found in 2–5% of all tuberous sclerosis patients with renal cysts [112,113]. These individuals have an increased likelihood of reaching ESKD from the ages of 20–30 years compared to ADPKD patients [114]. There is no clear evidence that treatment with tolvaptan is effective in these patients.

#### 3.6.2. Von Hippel-Lindau (VHL) Syndrome

Von Hippel-Lindau (VHL) syndrome is a hereditary systemic disease caused by a mutation in the *VHL* gene. This disease is inherited in an autosomal dominant manner and occurs in one in 50,000 individuals. The *VHL* gene is a tumor suppressive gene found on chromosome 3p25-26, and the 25–45% of VHL syndrome patients who have this gene mutation can also present with renal cell carcinoma, pancreatic cysts, central nervous system and retinal hemangioblastoma, and pheochromocytoma [115].

Clinical diagnostic criteria for VHL syndrome vary with family history; individuals with a clear family history are diagnosed by the presence of one VHL syndrome-related lesion, while individuals without a clear family history are diagnosed by the presence of multiple central nervous system angioblastomas or retinal hemangiomas, or by comorbidity of central nervous system angioblastoma or retinal hemangioma and VHL syndrome-related lesions [116] (Table 4).

Renal lesions associated with VHL syndrome broadly range from simple renal cysts to renal cell carcinoma and appear in 50–70% of patients with VHL syndrome [117,118,119]. These lesions are reported to appear in patients aged 20–30 years [116].

VHL syndrome is characterized by the frequent appearance of neoplastic or cystic lesions in multiple organs. In a study by Taylor et al. (2012), cysts were found in six out of 21 patients with VHL, four of whom had renal cysts [120]. Renal cysts in VHL do not produce any symptoms, and unlike ADPKD, kidney function impairment is rare. However, complex cysts may produce renal tumors. To determine the course of treatment, it is recommended to use the Bosniak classification (Table 2) in a regular image evaluation [17].

## 4. Summary

To help physicians evaluate cystic kidney diseases, this article provides a review of cystic kidney diseases for which a differential diagnosis is required for ADPKD. For each disease, we conduct a review and present image findings of specific cases that are representative of the disease. We believe this is a review paper that can be used in actual clinical practice.

## Figures and Tables

**Figure 1 jcm-11-06528-f001:**
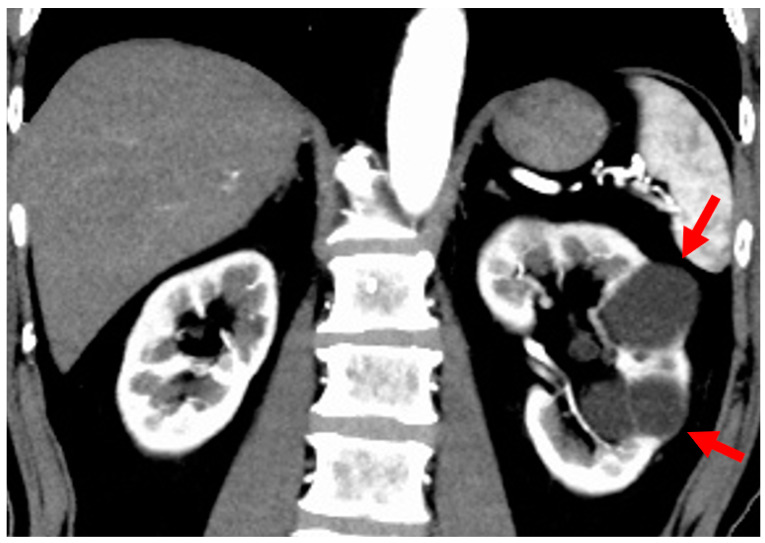
Multiple simple renal cysts: 60s, male, kidney function eGFR 68 mL/min/1.73 m^2^, Bosniak class I (image provided by Shonan Kamakura General Hospital). Several low-density cysts (arrow) on the left kidney.

**Figure 2 jcm-11-06528-f002:**
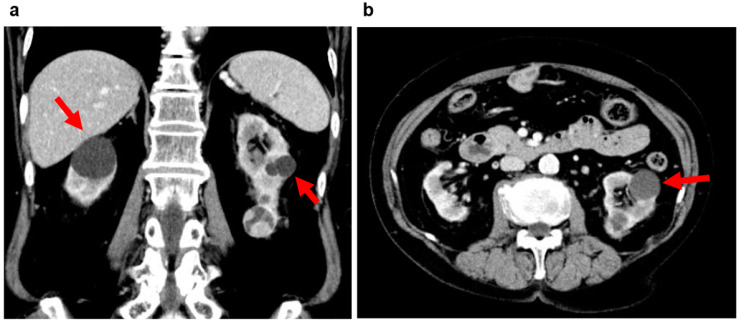
Acquired cystic kidney disease (ACKD): 70s, female, history of hemodialysis 3 years (primary disease: diabetic nephropathy), Bosniak class IV (image provided by Osaka Saiseikai Nakatsu Hospital). (**a**) Contrast CT image (coronal section): Several low-density cysts (arrow) in both kidneys and a densely stained solid area of cystic lesions is visible on the inferior pole of the left kidney. (**b**) Contrast CT image (horizontal section): Several low-density cysts (arrow) in both kidneys and a densely stained solid area of cystic lesions is visible on the inferior pole of the left kidney.

**Figure 3 jcm-11-06528-f003:**
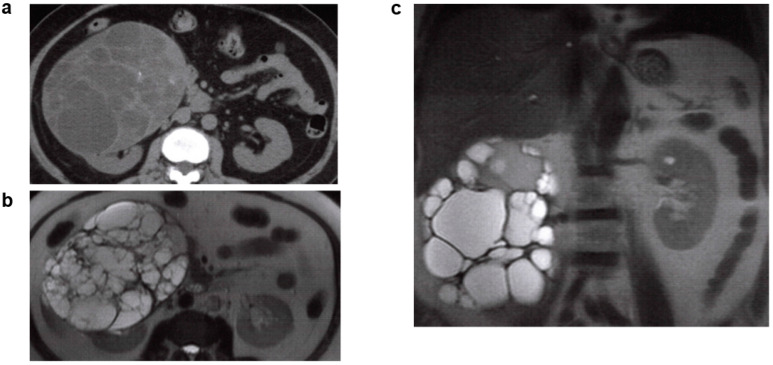
Multilocular renal cyst, multilocular cystic nephroma, polycystic nephroma: 60s, female, no family history of polycystic kidney disease or end-stage kidney disease, no history of hepatic cysts, cerebral aneurysm, or heart valve disease, kidney function eGFR 82.7 mL/min/1.73 m^2^ (image provided by Toranomon Hospital). (**a**) Non-contrast CT image (horizontal section): well-defined multilocular mass is visible on the right kidney. (**b**) MRI T2-weighted image (horizontal section): hyperintense polycystic mass (hypointense septum) is visible on the right kidney. (**c**) MRI T2-weighted image (coronal section): hyperintense polycystic mass (hypointense septum) is visible on the right kidney.

**Figure 4 jcm-11-06528-f004:**
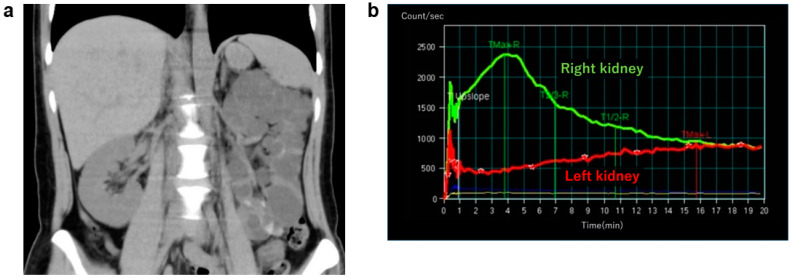
Multicystic kidney/multicystic dysplastic kidney (MCDK): 30s, female, no family history of polycystic kidney disease or end-stage kidney disease, hepatic lesions, cerebral aneurysm, or heart valve disease, kidney function eGFR 70.2 mL/min/1.73 m^2^ (images provided by Kurume University). (**a**) Non-contrast CT image (coronal section), unilateral (left side) several renal cysts were observed. Cyst’s density varied from low to high absorption. High density in hemorrhagic cysts, isodense areas suggest obsolete hemorrhage. Kidney volume was 222 mL for the right kidney, and 692 mL for the left kidney. (**b**) Renal dynamic scintigraphy: blood flow phase/functional phase/excretion phase of the right kidney (green line) remain good, indicating normal function. The 3 phases of the left kidney (red line) are deteriorating, indicating hypofunction.

**Figure 5 jcm-11-06528-f005:**
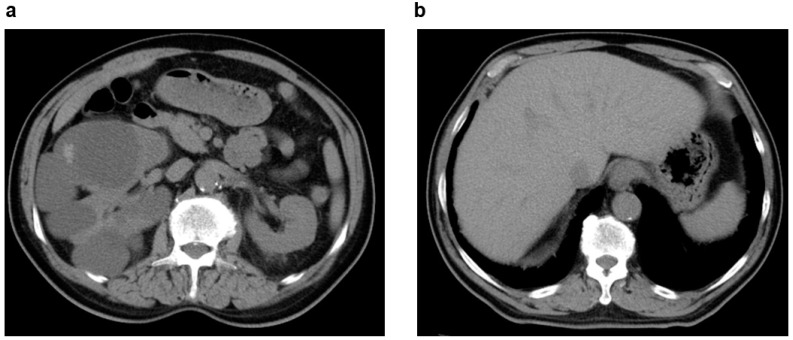
Unilateral renal cystic disease: 70s, male, no family history of polycystic kidney disease or end-stage kidney disease, no hepatic cysts, cerebral aneurysm, or history of heart valve disease, kidney function eGFR 82.7 mL/min/1.73 m^2^ (image provide by Kurume University). (**a**) Non-contrast CT image (horizontal section): the right kidney shows multiple variously sized cysts, while bilateral renal volume is 1401 mL. (**b**) Non-contrast CT image (horizontal section): no hepatic cysts.

**Figure 6 jcm-11-06528-f006:**
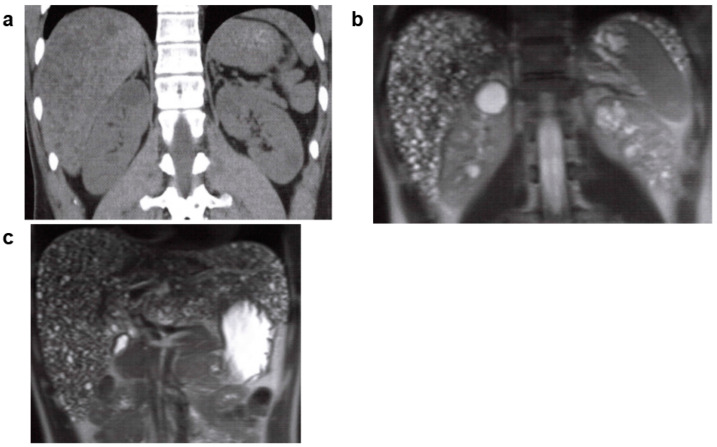
Autosomal recessive polycystic kidney disease (ARPKD): 30s, male, *PKHD1* gene complex heterozygous mutation (compound heterozygous mutation), older brother had polycystic kidney disease (PKD) with end-stage kidney disease, no other family history (including parents) of PKD or end-stage kidney disease, no history of cerebral aneurysm or heart valve disease, kidney function eGFR 25.4 mL/min/1.73 m^2^ (image provided by Toranomon Hospital). (**a**) Non-contrast CT image (coronal section): hepatomegaly with many microcysts of low density found on the liver. Multiple low-density cysts with a diameter of 1–3 cm are seen on the corticomedullary junction of both kidneys. (**b**) MRI T2-weighted image (coronal section): multiple hyperintense cysts with a diameter of 1–3 cm are seen near the corticomedullary junction of both kidneys. (**c**) MRI T2-weighted image (coronal section): multiple hyperintense microscopic hepatic cyst-like lesions are seen on the liver.

**Figure 7 jcm-11-06528-f007:**
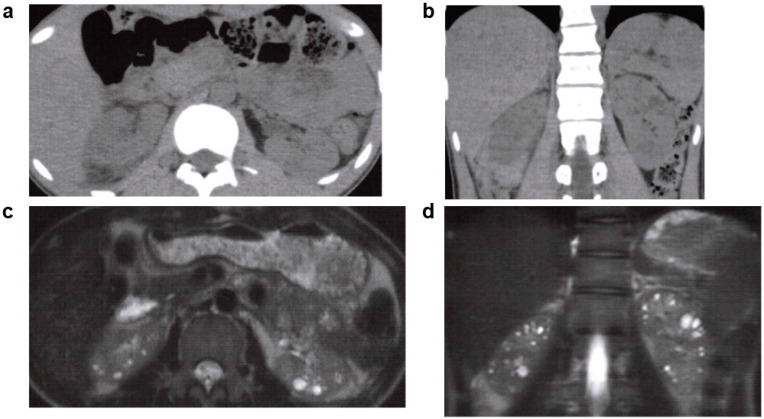
Autosomal dominant tubulointerstitial kidney disease (ADTKD): 20s, male, *UMOD* gene heterozygous mutation, positive for familial hyperuricemia, no family history of polycystic kidney disease or end-stage kidney disease, no history of cerebral aneurysm or heart valve disease, began dialysis at age 20 years (image provided by Toranomon Hospital). (**a**) Non-contrast CT image (horizontal section): many cysts in both kidneys, but there is no kidney enlargement. (**b**) Non-contrast CT image (coronal section): many cysts in both kidneys, but there is no kidney enlargement. (**c**) MRI T2-weighted image (horizontal section): many hyperintense cysts in both kidneys, but there is no kidney enlargement. (**d**) MRI T2-weighted image (coronal section): many hyperintense cysts in both kidneys, but there is no kidney enlargement.

**Figure 8 jcm-11-06528-f008:**
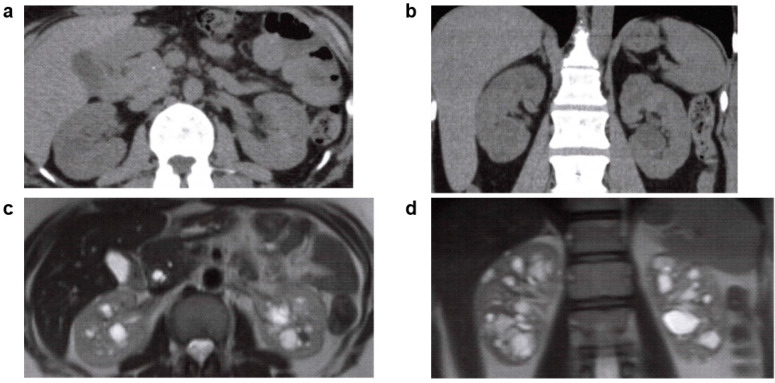
ADTKD: 40s: female, *HNF1B* gene heterozygous mutation, no family history of polycystic kidney disease or end-stage kidney disease, no history of cerebral aneurysm or heart valve disease, kidney function eGFR 39.2 mL/min/1.73 m^2^ (image provided by Toranomon Hospital). (**a**) Non-contrast CT image (horizontal section): multiple cysts in both kidneys, but there is no kidney enlargement. (**b**) Non-contrast CT image (coronal section): multiple cysts in both kidneys, but there is no kidney enlargement. (**c**) MRI T2-weighted image (horizontal section): multiple hyperintense cysts of various sizes are noted near the corticomedullary junction on both kidneys, but there is no kidney enlargement. (**d**) MRI T2-weighted image (coronal section): multiple hyperintense cysts of various sizes are noted near the corticomedullary junction on both kidneys, but there is no kidney enlargement.

**Figure 9 jcm-11-06528-f009:**
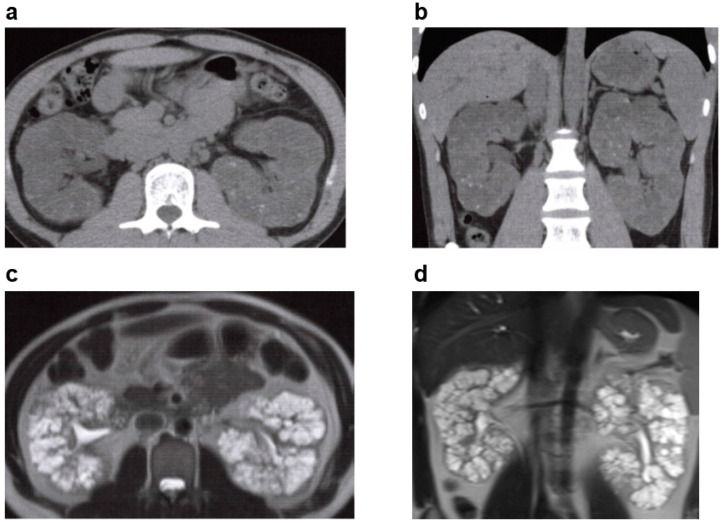
Nephronophthisis (NPH): 20s, male, *NPHP4* gene compound heterozygous mutation, no family history of polycystic kidney disease or end-stage kidney disease, no history of cerebral aneurysm or heart valve disease, kidney function eGFR 29.4 mL/min/1.73 m^2^ (image provided by Toranomon Hospital). (**a**) Non-contrast CT image (horizontal section): multiple low-density regions in both kidneys. (**b**) Non-contrast CT image (coronal section): multiple low-absorbance regions in both kidneys. (**c**) MRI T2-weighted image (horizontal section): multiple hyperintense cyst-like lesions in both kidneys. (**d**) MRI T2-weighted image (coronal section): multiple hyperintense cyst-like lesions in both kidneys.

**Figure 10 jcm-11-06528-f010:**
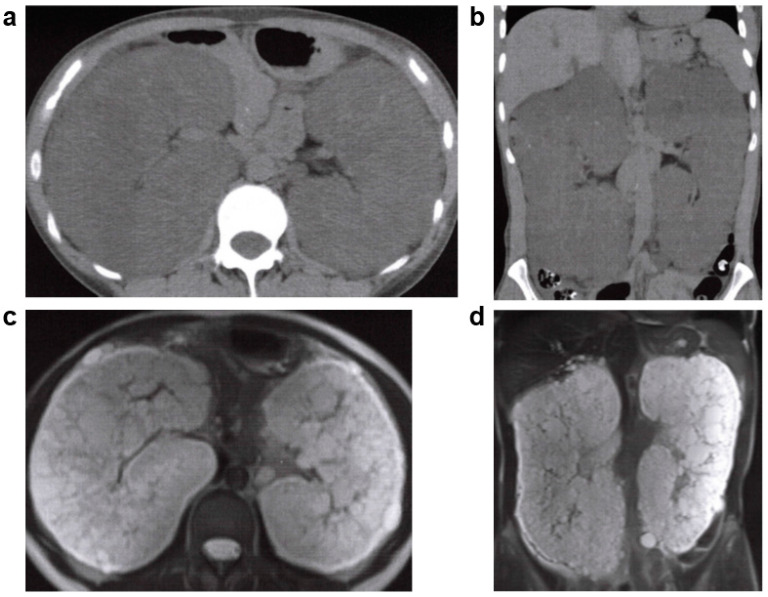
Oral-facial-digital syndrome (OFD) type 1: 30s, female, *OFD1* gene heterozygous mutation, no family history of polycystic kidney disease or end-stage kidney disease, no history of hepatic cysts, cerebral aneurysm, or heart valve disease, no abnormalities in urinalysis, began dialysis at age 26 years (image provided by Toranomon Hospital). (**a**) Non-contrast CT image (horizontal section): multiple low-density cystic lesions are in both kidneys, accompanied by kidney enlargement. There is no change in the external appearance of the kidneys. (**b**) Non-contrast CT image (coronal section): multiple low-density cystic lesions are in both kidneys, accompanied by kidney enlargement. There is no change in the external appearance of the kidneys. (**c**) MRI T2-weighted image (horizontal section): multiple hyperintense cystic lesions are in both kidneys, accompanied by kidney enlargement. There is no change in the external appearance of the kidneys. (**d**) MRI T2-weighted image (coronal section): multiple hyperintense cystic lesions are in both kidneys, accompanied by kidney enlargement. There is no change in the external appearance of the kidneys.

**Figure 11 jcm-11-06528-f011:**
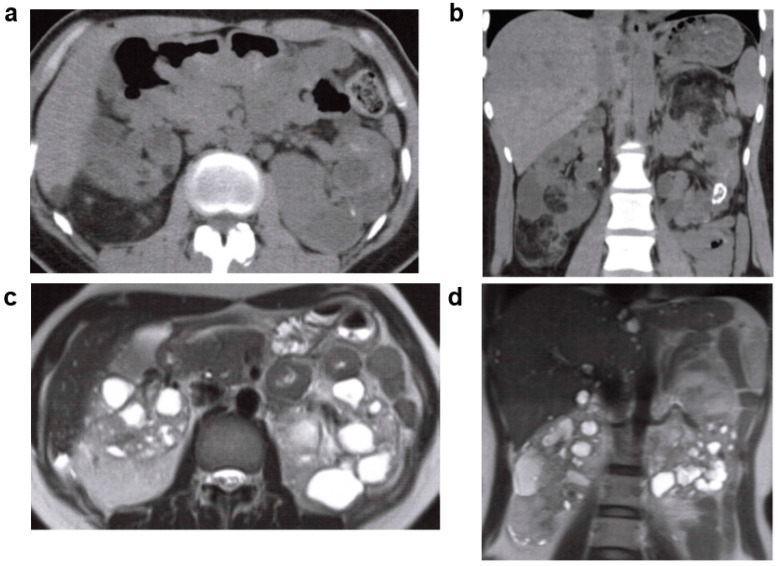
Tuberous sclerosis: 30s, female, wide-ranging loss of heterozygosity noted on the *TSC2* gene, no family history of kidney disease, positive for hypopigmented macule/facial angiofibroma/pulmonary lymphangioleiomyomatosis, positive for epilepsy history, kidney function eGFR 55.3 mL/min/1.73 m^2^ (image provided by Toranomon Hospital). (**a**) Non-contrast CT image (horizontal section): multiple lipomas and low-density cysts in both kidneys. (**b**) Non-contrast CT image (coronal section): multiple lipomas and low-density cysts in both kidneys. (**c**) MRI T2-weighted image (horizontal section): multiple hyperintense cysts of varying sizes in both kidneys. (**d**) MRI T2-weighted image (coronal section): multiple hyperintense cysts of varying sizes in both kidneys.

**Table 1 jcm-11-06528-t001:** Summary of cystic kidney disease.

Disease	Gene	Inheritance	Prevalence	Clinical Features
ADPKD	*PKD1*, *PKD2*	AD	1/1000–1/2500	Proliferative cysts in both kidneys increase in number and size as kidney function impairment advances, often introducing complications such as hypertension, multiple hepatic cysts, cerebral aneurysm, heart valve disease, and colonic diverticula.
Multiple simple renal cysts			Common	Usually a simple cyst of only unilateral, but with age the number of cysts increases in both kidneys. Normally asymptomatic.
ACKD			Common	In long-term dialysis patients, the renal parenchyma in both kidneys may show atrophy and proliferation of microcysts. Asymptomatic, but screening for hemorrhage and renal cancer development is necessary.
MCDK			1/1000–1/4300	Renal cysts of various sizes gather. Renal prognosis is favorable if no complications such as VUR in the contralateral kidney.
URCD			Unknown	Unilateral renal cysts resembling ADPKD in imaging. No family history of PKD, no progress to ESKD, and no complications such as hepatic cysts or cerebral aneurysm.
ARPKD	*PKHD1*, *DZIP1L*	AR	1/8000–1/40,000	Renal cystic lesions reflecting the expansion of the collecting duct with kidney function impairment, and congenital hepatic fibrosis characterized by bile duct dysplasia and intrahepatic periportal fibrogenesis. Thrombocytopenia, splenomegaly, and portal hypertension.
ADTKD	*UMOD*, *MUC1*, *REN*, *HNF1B*, *SEC61A*, *DNAJB11*	AD	Unknown	Tubulointerstitial fibrogenesis and progressive kidney function impairment. Poor urinary findings, slow decline in kidney function. Small renal cysts without kidney enlargement. ADTKD-*UMOD*, *-MUC1*, *-REN*, *-SEC61A*; Hyperuricemia and Gout, ADTKD-*HNF1B*; diabetes mellitus and liver function impairment, ADTKD-*SEC61A*; congenital anemia, neutropenia, and hypogammaglobulinemia. ADTKD-*DNAJB11*; circulatory abnormalities such as intracranial aneurysm, thoracic aorta enlargement, and carotid artery dissociation.
NPH	*NPHP1*-*20*, *NPHP1L*, *NPHP2L*, *TRAF31P1*, *AH11*, *CC2D2A*	AR	1/50,000–1/100,000	Impaired urinary concentration, chronic tubulointerstitial nephritis, renal cystic lesions and accompanying kidney function impairment and progress to ESKD by the age of 30 years. Extrarenal lesions are reportedly found in 10–20% of patients, retinitis pigmentosa, cerebellar vermis hypoplasia, gaze palsy, hepatic fibrosis, and skeletal abnormalities.
OFD type1	*OFD1*	X-linked	1/50,000–1/250,000	Progress to kidney failure in adulthood with PKD with a variety of morphological abnormalities in the oral, facial, and finger and toe. Normal to large-sized kidneys.
TSC	*TSC1*, *TSC2*	AD	1/10,000	Hamartoma in the skin, nervous system, kidney, lung, bone, and elsewhere. Renal lesions; angiomyolipoma (AML), renal cysts, renal cell carcinoma.
VHL syndrome	*VHL*	AD	1/50,000	Renal cell carcinoma, pancreatic cysts, central nervous system and retinal hemangioblastoma, and pheochromocytoma. Kidney function impairment is rare.

Abbreviations: autosomal dominant polycystic kidney disease (ADPKD), acquired cystic kidney disease (ACKD), multicystic dysplastic kidney (MCDK), unilateral renal cystic disease (URCD), autosomal recessive polycystic kidney disease (ARPKD), autosomal dominant tubulointerstitial kidney disease (ADTKD), nephronophthisis (NPH), oral-facial-digital syndrome (OFD), tuberous sclerosis (TSC), Von Hippel-Lindau (VHL), autosomal dominant (AD), autosomal recessive (AR), polycystic kidney disease (PKD), end-stage kidney disease (ESKD).

**Table 2 jcm-11-06528-t002:** Bosniak Classification [17].

Class	
I	Hairline-thin wall; water attenuation; no septa, calcifications, or solid components; nonenhancing
II	Two types: 1. Few thin septa with or without perceived (not measurable) enhancement; fine calcification or a short segment of slightly thickened calcification in the wall or septa 2. Homogeneously high-attenuating masses ≤ 3 cm that are sharply marginated and do not enhance
IIF	Two types: 1. Minimally thickened or more than a few thin septa with or without perceived (not measurable) enhancement that may have thick or nodular calcification 2. Intrarenal nonenhancing hyperattenuating renal masses ≥ 3 cm
III	Thickened or irregular walls or septa with measurable enhancement
IV	Soft-tissue components (i.e., nodule[s]) with measurable enhancement

**Table 3 jcm-11-06528-t003:** Diagnostic criteria for tuberous sclerosis [105].

Major Features	Minor Features
1. Hypomelanotic macules (≥3, at least 5-mm diameter) 2. Angiofibromas (≥3) or fibrous cephalic plaque 3. Ungual fibromas (≥2) 4. Shagreen patch 5. Multiple retinal hamartomas 6. Cortical dysplasias 7. Subependymal nodules 8. Subependymal giant cell astrocytoma 9. Cardiac rhabdomyoma 10. Lymphangioleiomyomatosis (LAM) 11. Angiomyolipomas (AML) (≥2)	“Confetti” skin lesions Dental enamel pits (≥3) Intraoral fibromas (≥2) Retinal achromic patch Multiple renal cysts Nonrenal hamartomas
Definitive diagnosis: 2 major features or 1 major feature with 2 minor features. Alternatively, the identification of either a *TSC1* or *TSC2* pathogenic mutation.	
Possible diagnosis: Either 1 major feature or ≥2 minor features	

**Table 4 jcm-11-06528-t004:** Diagnostic criteria for Von Hippel Lindau syndrome [116].

(1) With a positive family history of VHL syndrome
Retinal angiomas, Spinal or cerebellar hemangioblastomas, Endolymphatic sac tumors, Renal cell carcinoma, Pheochromocytoma, Pancreatic lesions (Pancreatic cysts, Pancreatic neuroendocrine tumors), Epididymal cystadenomas, etc.
(2) With no clear family history of VHL syndrome
1. Spinal or cerebellar hemangioblastomas or Retinal angiomas (≥2) 2. Spinal or cerebellar hemangioblastomas or Retinal angiomas (≥1) along with 1 of the following: a. Renal cell carcinoma, b. pheochromocytoma, c. Pancreatic cyst or Pancreatic neuroendocrine tumors, d. Epididymal cystadenomas, e. Endolymphatic sac tumors 3. 1 of the above clinical features with identification of a heterozygous germline pathogenic mutations in *VHL* on molecular genetic testing

## Data Availability

The data that support the findings of this study are available from the corresponding author, upon reasonable request.

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
