# Peer review of "Cystic Kidney Diseases That Require a Differential Diagnosis from Autosomal Dominant Polycystic Kidney Disease (ADPKD)"

_jcm, 2022, doi:10.3390/jcm11216528_

Round 1

Author Response

14-October-2022

Dr. Emmanuel Andres,

Editor-in-Chief,

Journal of Clinical Medicine,

Dear Dr. Andres

We would like to thank you for the letter dated 10/11 /2022, and the opportunity to resubmit a revised manuscript. We would also like to take this opportunity to express our thanks to the reviewers for the positive feedback and helpful comments for correction or modification.

We believe have resulted in an improved revised manuscript, which you will find upload alongside this document. The manuscript has been revised to address the reviewer comments, which are appended alongside our responses to this letter.

Thank you very much for reviewing our manuscript and offering valuable advice.

We have addressed your comments with point-by-point responses below and revised the manuscript accordingly.

-------------------------

Reviewer 1:

Overall, this review paper is well written and provides pertinent information about cystic kidney diseases and their clinical manifestation and diagnostic criteria to the reader - it can, indeed, be a useful source of information to clinicians. The cited references seem overall up to date, English language and style are correct. I was, however, somewhat surprised that symptoms/manifestations that alone or as a sum allow to differentiate the described diseases from ADPKD are sometimes, but not systematically, presented for each disease. In this context I would suggest the inclusion of a summary table allowing an overview of ADPKD in comparison to the other presented cystic kidney diseases and including differentiating symptoms (this could be similar to Table 2 presented in Alves M, Fonseca T, de Almeida EAF. Differential Diagnosis of Autosomal Dominant Polycystic Kidney Disease. In: Li X, ed. Polycystic Kidney Disease. Brisbane (AU): Codon Publications; November 2015).

Thank you for this suggestive comment. We added Table 1.

[74-87 Introduction]: I think that a more detailed clinical description of ADPKD (TKV increase, kidney function abnormalities and decline, etc.), including applied diagnostic criteria (cyst number, underlying pathogenic variants not affecting the PKD1 and PKD2 genes, etc.) as well as the most prevalent extrarenal manifestations (e.g. hypertension, vascular and cardiac manifestations, polycystic liver disease and further extrarenal cysts and lesions) would be beneficial to the reader of this review article. Indeed, I was somewhat surprised that most of the other described cystic kidney diseases seem to be presented in more detail than ADPKD itself.

Thank you for your suggestion. We added the following text: “affecting one in 1,000-2,500 individuals and is mainly caused by mutations in the PKD1(78%) and PKD2(15%) genes [1]. Approximately 7% of patients with ADPKD appear not to have a PKD1/2 mutation [2, 3]. Among ADPKD patients, the severity of the effects of these mutations on renal survival was reported to be as follows: PKD1 truncating mutation > PKD1 non-truncating mutation > PKD2 mutation > no PKD1 or PKD2 mutation [4, 5].” (line 75-80)

[208-212, 360-371]: Are there specific HNF-1B mutations leading to either MCDK or ADTKD? Could you maybe elaborate on this?

Thank you for pointing this out. HNF1B gene mutations leads to CAKUT, MCDK, and ADTKD. We changed the sentence “MCDK was thought to be a nonhereditary renal disease, but in recent years, a human HNF1B gene mutation was found to have a causal relationship with a renal developmental defect accounting for 10–30% of all cases of kidney and urinary tract congenital abnormalities (CAKUT) [40], MCDK, and  autosomal dominant tubulointerstitial kidney disease (ADTKD) [41]” (line 218-223)

[214-227]: In the beginning of the paragraph the authors cite that URCD does not progress to ESRD, while at the end of the paragraph the authors outline the necessity of regular imaging of patients presenting ESRD in order to monitor the possible formation of renal cell carcinomas in the context of acquired cystic kidney disease (ACKD) due to dialysis/CKD. To me it appears a little confusing to mix the notions of URCD and ACKD without further explanation. I think it would be beneficial to clarify this point and introduce the cited reference (Sun et al. 2018) in the paragraph dedicated to ACKD [125-163].

Thank you for your suggestion. We've removed the sentence “In patients with ESRD, cysts proliferate unilaterally or bilaterally in atrophic kidney, and regular imaging is necessary as renal cell carcinoma is a possible complication.”, because the sentence was referring to ACKD.

[273-278]: I think that it would be more coherent to add this paragraph about ADPKD treatment to a more detailed introduction on ADPKD rather than leave it in the section treating ARPKD. Furthermore, the formulation “no established disease-specific treatment exists” for ADPKD is somewhat misleading as Tolvaptan, which the authors cite in line 537, is specifically used to reduce renal cyst growth in fast progressing ADPKD (in eligible patients presenting no serious hepatic affections) and helps retard ESRD and kidney cyst growth without positively affecting other systemic manifestations of ADPKD. I think it would be beneficial to the paper to clarify this point.

Thank you for your suggestion. Actually, where I write ARPKD, we miswrote it as ADPKD. We changed the word accordingly. We added the following text: “The only therapeutic drug for patients with ADPKD is tolvaptan, a vasopressin V2-receptor antagonist that slows the rate of decline of the estimated glomerular filtration rate (eGFR) [8-10]. Accordingly, it is important to diagnose ADPKD from other cystic kidney disease.” (line 90-93)

[208-212, 284, 360-371]: I would propose to use only one gene symbol for HNF-1B instead of using different names, which could confuse readers.

Thank you for your suggestion. We unified the description of the gene to HNF1B. (line 219, 372-374, 380)

[413, 441]: “Two to three of all NPH cases […]” (line 413) – sentence not clear. Do the authors mean 2-3% or two thirds as mentioned in line 441 “[…] two thirds of all cases have no clear gene mutation […]”?

Thank you for your suggestion. We modified that expression from “Two to three” to “Two-thirds”. (line 425)

-------------------------

Reviewer 2:

  1. Three tables are mentioned in the main text, but they are not provided in the manuscript. The tables should be included in the revised manuscript.

Thank you for pointing this out. We added the tables.

  1. The medical images presented in this article may be easy to understand for a nephrologist, but they are difficult to understand for a broad audience. Some annotationand labeling of images for disease features may help a broad audience understand the diagnostic criteria.

Thank you for pointing this out. We added descriptions of arrows pointing to the cysts in Fig1 and Fig2.

  1. The style for references should be used in a unified format. For example, the style of references 2, 10, 22, 25-27, 29, 40, 44, 55-56, 65, 67, 70-71, 74, 79-80, 82, 85-86, 104 should be modified.

Thank you for your suggestion. We modified the “references” section.

  1. In line 96, Prevalence rates can vary depending on age, gender, and ethnicity 567.The reference 567 is mislabeled.Three tables are mentioned in the main text, but they are not provided in the manuscript. The tables should be included in the revised manuscript.

Thank you for your suggestion. We modified the label. (line 105)

  1. In lines 153, & 155, “Ishikawa et al (2011). reported“, & “Kitajima et al (2019). reported”, please remove the period or please modify it as “Ishikawa et al. (2011) reported“, & “Kitajima et al. (2019) reported”,

Thank you for your suggestion. We I changed those sentences correctly; “Ishikawa et al. (2011) reported…” (line 163) and “Kitajima et al. (2019) reported” (line 165).

We made some corrections throughout the manuscript.

We added (URCD) (line 62), (OFD) (line 66), (TSC) (line 67), and (VHL) syndrome (line 67, 556, and 559).

We correct the sentence” Usually a simple cyst of only unilateral, but with age the number of cysts increases and the frequency with which they are present in both kidneys increases[11]. Simple renal cysts are rarely found in childhood but increase with age after the age of 30. Autopsy reports indicate the presence of at least one renal cyst in 50% of patients aged 50 years or above.” (line 100-104)

We corrected the word “kidney function impairment”, “kidney function”, “kidney failure”, “kidney transplantation”, “kidney replacement”, “kidney enlargement”, “end-stage kidney disease (ESKD)”, “hemodialysis”, “liver function impairment”, “tubulointerstitial damage” (line 110, 112-3, 146, 172, 205, 214, 229, 262, 285, 286, 287, 293, 299, 300, 301, 307, 324, 343, 345, 378, 383, 401, 402, 411, 431, 459, 460, 461, 488, 489, 515, 516, 535, and 571)

We corrected the word “CT” to “computed tomography (CT)” (line 122)

We correct the sentence “Autosomal recessive polycystic kidney disease (ARPKD) presents as renal cystic lesions reflecting the expansion of the collecting duct with kidney function impairment, and congenital hepatic fibrosis characterized by bile duct dysplasia and intrahepatic periportal fibrogenesis[49].” (line 240-3)

We correct the sentence “The majority of these patients showed multiple bilateral cysts (³10 cysts per kidney) on ultrasound, with half cases reported to be CKD stage 1–3.” (line 248-50)

We correct the sentence “Extrarenal diseases include juvenile onset diabetes mellitus and liver function impairment, as well as abnormalities of the reproductive organs[68].” (line 377-8)

We correct the sentence “Clinical diagnostic criteria for VHL syndrome varies with family history; individuals with a clear family history are diagnosed by presence of one VHL syndrome-related lesion, while individuals without a clear family history are diagnosed by the presence of multiple central nervous system angioblastomas or retinal hemangiomas, or by comorbidity of central nervous system angioblastoma or retinal hemangioma and VHL syndrome-related lesions[117] (Table 4).” (line 560-4)

We correct the sentence “However, complex cysts may produce renal tumors. To determine the course of treatment, it is recommended to use the Bosniak classification (Table 2) in a regular image evaluation[17].” (line 572-3)

We correct “Author Contributions” section.

Sincerely,

Satoru Muto, MD, PhD

Department of Urology, Juntendo University Nerima Hospital

3-1-10, Takanodai, Nerima-ku, Tokyo 177-8521, Japan

Tel: +81-3-5923-3111, Fax: +81-3-5923-3217

E-mail: s-muto@juntendo.ac.jp

Reviewer 2 Report

The article by Sekine et al. present a review of the differential diagnosis of the cystic kidney diseases to validate autosomal dominant polycystic kidney disease (ADPKD) and other cystic kidney diseases. This article is well written and provides high quality medical imaging to facilitate the diagnosis of different cystic kidney diseases. But a few issues should be improved to allow me to accept their articles for publication.

1.      Three tables are mentioned in the main text, but they are not provided in the manuscript. The tables should be included in the revised manuscript.

2.      The medical images presented in this article may be easy to understand for a nephrologist, but they are difficult to understand for a broad audience. Some annotation and labeling of images for disease features may help a broad audience understand the diagnostic criteria.

3.      The style for references should be used in a unified format. For example, the style of references 2, 10, 22, 25-27, 29, 40, 44, 55-56, 65, 67, 70-71, 74, 79-80, 82, 85-86, 104 should be modified.

4.      In line 96, Prevalence rates can vary depending on age, gender, and ethnicity 567.The reference 567 is mislabeled.

5.      In lines 153, & 155, “Ishikawa et al (2011). reported“, & “Kitajima et al (2019). reported”, please remove the period or please modify it as “Ishikawa et al. (2011) reported“, & “Kitajima et al. (2019) reported”,

Author Response

(The authors gave the same response as above.)
